# Hybrid Composite Materials Made of Recycled PET and Standard Polymer Blends Used in Civil Engineering

**DOI:** 10.3390/polym15163407

**Published:** 2023-08-14

**Authors:** Daniel Papán, Lenka Lapašová, Zuzana Papánová

**Affiliations:** 1Department of Structural Mechanics and Applied Mathematics, Faculty of Civil Engineering, University of Žilina, Univerzitná 8215/1, 010 26 Žilina, Slovakia; 2Faculty of Civil Engineering, University of Žilina, Univerzitná 8215/1, 010 26 Žilina, Slovakia

**Keywords:** cement–concrete, polymer composite, mechanical properties, stress–strain diagram, force stand, modern material, hybrid composite, polyethylene tetraphthalate

## Abstract

This paper presents an investigation of the tensile properties of two composites made from recycled polyethylene tetraphthalate, cement–concrete mix and standard polymer-based adhesive mixes, used in the construction industry. To describe tensile behavior, experimental measurements of each component of the resulting composite materials were processed in detail. It presents the possibilities of modifying materials suitable for building structures and at the same time provides an opportunity to get rid of polyethylene tetraphthalate (PET), which has already been recycled several times. Because the resulting composite contains a majority of the composite composition on a fragile basis, its use in practice depends on its simple thrust properties. In this paper, a study of the most important mechanical properties of a previously unused composite is presented. These properties were obtained experimentally using an innovative tensile test method.

## 1. Introduction

It is common knowledge that concrete belongs to the most widely used building materials. It can be defined as a composite material formed by hardening a mixture of its basic components (aggregate, water and the most used binder—cement). Currently, other types of binders are also used, such as asphalt (asphalt concrete) [1], polymers (polymer concretes) [2] or sulfur (sulfuric concretes) [3]. The properties of concrete are mainly influenced by its composition and the ratio of its basic components. 

In this experimental work, plain concrete without steel-reinforcing bars and wires with static functions are used. This type of concrete is characterized by high compressive strength but low tensile strength. Concrete reinforcement is necessary to precisely improve the concrete’s tensile strength. The most common methods of reinforcement are steel-reinforcing bars or wire. These favorably affect the bearing capacity in the tensile and shear zones of the element. Fiber reinforcement is a modern technique to strengthen concrete. This type of reinforcement is especially advantageous when reinforcing more complicated and thin elements where it is impossible to safely and correctly store ordinary arming reinforcement [4,5,6]. Fiber-reinforced concrete (FRC) properties are influenced primarily by the type of concrete and the geometry, distribution, orientation and density of the fibers.

This experiment focuses on the effective use of recycled polyethylene terephthalate (PET) in concrete because of the worldwide issue regarding its processing. Waste PET material is one of the most important and widespread in the world. PET is mainly used to produce beverage packaging. Current global PET production exceeds 6.7 million tons per year [7]. One possible solution for the use of recycled PET is precisely short reinforcing fibers [4]. Plastic fibers have been found to have several advantages over more traditional steel fibers in the past. These include significantly lower weight for the same volume, lower transport costs, higher corrosion resistance, high impermeability of FRC, higher compatibility with admixtures, lower thermal conductivity and higher workability. In a hybrid composite, we can optimize the interaction between the individual components by enveloping polyethylene terephthalate (PET) reinforcement fibers with reactoplastic.

Hybrid composites are defined as systems in which one type of reinforcing material or filler is included in mixtures of different matrices [8]. In this case, two variants of test composites were prepared. These were the composite specimen using a vinylester anchor (Figure 1) and the composite specimen using a polyester resin. The structure of the composite will depend on its intended use. The main requirement in improving materials used in construction is the development of new composite materials with increased strength and durability compared to traditional types. The next sections describe the properties of the components of the composite.

### Significance of this Research

The significance of this research is in the previously unexplored composition of the materials. Their interconnection combines the ecological importance and economic utility of polymers in view of the enormous worldwide production of PET materials. It is one of the most efficient ways of using thermoplastics that are no longer recyclable. Furthermore, the properties of two differently hardening bonding materials are investigated: this hardening takes place on different chemical bases.

Compared to research around the world, this type of material is being studied for a different application. The research presented here addresses the field of application in road construction. The research presented is on tensile stress, which is the basis of application for this purpose. A lot of research is devoted to standard cement–concrete and difficult-to-manufacture reinforcing fibers. The hybrid composite presented in the paper also has the advantage of ease of production of both binders and PET particles.

In the conclusion of the article, there are also references to studies that present research on similar topics.

## 2. Materials of the Tested Composites

### 2.1. Cement–Concrete Mixture

As mentioned already, cement–concrete is the most used building material, having high compressive strength but low tensile strength. The compressive strength of concrete ranges from 5 to 50 MPa, but in tensile strength, it reaches only about a tenth of the value of its compressive strength. The filler fraction is very important, as it affects the strength of concrete. The overall properties of cement–concrete depend primarily on the composition and processing of the mixture. It is resistant to chemicals, high temperatures, durable and has volumetric stability. Its durability is great both in air and in water.

### 2.2. Reactoplastics

#### 2.2.1. Vinylester Anchor

This is a two-component chemical anchoring system based on vinylester resins with a short curing time. It works on the principle of the high reactivity of unsaturated vinylester resins in methacrylate monomers. Such a resin forms a strong and, at the same time, chemically resistant joint. The advantage is also the reduced hardening time and easy application, which is the main criterion in the design of the hybrid composite dealt with in this research work. The tensile strength of the vinylester anchor is approximately 13–14 MPa [9,10,11].

#### 2.2.2. Polyester Resin

Based on the chemical nature of the polymer main chain, polyester resins can be classified into saturated and unsaturated forms. The main difference between saturated and unsaturated polyester resins is the existence of double bonds in the main chain. Unsaturated polyester resin has them in its main chain, whereas saturated polyester resin does not [12]. Polyester resin has a relatively wide scope of application. It has excellent properties, such as resistance to chemicals, very high strength after complete hardening and other advantages.

### 2.3. Recycled PET

PET is currently the most used polymer material (thermoplastic) for the packaging of various products [4]. In general, PET is known for its high strength, durability, resistance to damage and non-biodegradability. Its fibers or particles are used as fibercrete components to achieve excellent properties.

One of the process characteristics is the melt flow rate. The MFR (melt flow rate) is a parameter used to characterize the viscosity and flowability of plastic materials, including recycled PET. It is expressed in units of grams per 10 min (g/10 min). It measures the rate at which a polymer is able to pass through a calibrated orifice under standardized conditions.

Most of the non-alcoholic PET bottles used in Slovakia are made of PET material type KOSA PET 1101 (*M_n_* = 27 kg/mol, *M_w_* = 65 kg/mol, density 1168 g/cm^3^).

The melt flow rate (MFR) was measured according to ISO 1133. At a load of 2.16 kg and 280 °C, the MFR is 33.8/10 min.

Fibers obtained from PET bottles can be used in the concrete mixture at a high rate (up to 3% by weight of cement). Their density is 1.41 g/cm^3^ and the elasticity module is 1700 MPa. Tensile strength ranges from 25 to 34 MPa [13,14,15].

### 2.4. Reinforcing Fibers

It is important to note that reinforcing fibers do not improve tensile strength but regulate cracks. This increases the ability to withstand the reinforced elements. Some types of fibers used in the manufacture of concrete have very good resistance to impact, abrasion and total destruction. A longer steel or synthetic fiber can replace concrete reinforcement in certain situations [16].

The polymer fibers were found to increase the splitting tensile strength of all types of FRC. The greatest increase in splitting tensile strength (up to 25%) was achieved when using PET fiber (length 80 mm) [17].

The amount of PET waste continues to grow as PET production and consumption increase. PET waste has become a major problem of environmental pollution. The produced waste is discarded by burial or incineration. However, fiber concrete can use it. PET fibers can be obtained from PET waste by a simple cutting method without any chemical treatment.

## 3. Methods

### 3.1. Static Tensile Test

Tensile testing is used to measure the force required to break a composite specimen and to determine the extent to which the specimen will elongate to the point of breakage. These tests are important to generate a stress–strain diagram. The latter is used to determine the modulus of tensile strength.

The mechanical behavior of materials is described by their deformation and refraction characteristics (tensile, compression, multi-axis stress). Mechanical behavior depends on several variables, including material composition, testing methods and the nature of the stresses generated [18]. Since the physical properties of many materials can vary depending on the ambient temperature, it is therefore appropriate to test the materials at temperatures that simulate the end-use environment.

Tensile response is influenced by various factors such as type of material, construction technology, compaction and fixing of specimens, testing speed and others. The tensile test is used to obtain basic information on the strength of materials and as an acceptance test for material specifications. The specimen is loaded with a uniaxial force *F*, and it is tested for elongation until it breaks (Figure 2).

From the elongation measurements made, a curve of dependence between the stress and relative strain is constructed. The stress *σ* from this dependence curve is evaluated as the ratio of the applied force *F* and the original cross-sectional area of the test specimen *A*_0_ [19].
(1)$$\sigma =\frac{F}{A_{0}}\;\left[MPa\right]$$

The deformation in the *x*-axis direction ε used in the stress–strain diagram is calculated as the ratio of the change in the length of the specimen Δ*L* and its original length *L*_0_.
(2)$$\varepsilon =\frac{\Delta L}{L_{0}}=\frac{L-L_{0}}{L_{0}}\left[-\right]$$

Determining the tensile strength is relatively simple. The test is repeatable. This material property is applicable for product quality control. Empirical relationships based on the correlation between tensile strength and properties such as hardness and fatigue resistance are often used in practice. For fragile materials, tensile strength is a valid design criterion. Tensile strength *f_t_* is the maximum load *F_max_* divided by the cross-sectional area *A* of the specimen (3).
(3)$$f_{t}=\frac{F_{max}}{A}[MPa]$$

The advantage of this type of solution is the independence of the stress–strain curve from the dimensions of the specimen.

A tearing machine or tensile testing machine is used to do this type of test. Together with other additional and auxiliary equipment, it is one of the basic equipment of every testing laboratory.

### 3.2. Constitutive Material Models

Regression analysis is primarily used to analyze interrelationships between variables. Its task is to find the mathematical regression function that best describes the course of dependence between variables. Analyzing the relationship between the dependent variable *y* and the set of independent variables *x* is the objective of this analysis (16). This application is found in various areas.

According to the number of unknowns variables, we divide the regression analysis into

Hyperbolic regression analysis;Cubic regression analysis.

#### 3.2.1. Hyperbolic Regression Analysis

The parameters of hyperbolic regression *a*, *b* are determined by the least-squares method (17). The use of this method leads to the following mathematical equations:(4)Equation: y^=a+bx
(5)Coefficient b: b=n∑yixi−∑1xi−∑yin∑1xi2−(∑1xi)2
(6)Coefficient a: a=1n∑yi−bn∑1xi
(7)Standard error of the regression: A¯=1n∑yi−y^iyⅈ⋅100%

#### 3.2.2. Cubic Regression Analysis

Cubic regression analysis can be used to calculate the coefficients *a*, *b*, *c* and *d* based on the following mathematical relationships:(8)$$\hat{y}=ax^{3}+bx^{2}+cx+d$$

A system of equations to find *a*, *b*, *c*, *d*:a∑xi3+b∑xi2+c∑xi+nd=∑yi,
a∑xi4+b∑xi3+c∑xi2+d∑xi=∑xiyi,
a∑xi5+b∑xi4+c∑xi+d∑xi2=∑xi2yi
(9)a∑xi6+b∑xi5+c∑xi4+d∑xi3=∑xi3yi
(10)Correlation coefficient: R=1−∑(yi−yi^)2∑(yi−y¯)2
(11)y¯=1n∑yi
(12)Standard error of the regression: A¯=1n∑yi−y^iyⅈ⋅100%

The display of the individual regression curves is shown in Figure 3. The red curve shows the behavior of the hyperelastic material (PET). The blue curve approximates thermosets (polyester resin, vinylester). The green curve on a certain part represents the behavior of cement–concrete.

## 4. Experiment

### 4.1. Test Specimen

The biggest problem in testing is gripping the specimen. It is necessary to ensure that the specimen does not move in the gripping area under maximum load. A typical specimen for a uniaxial tensile test is shown in Figure 2.

The measuring part is one of the most important components of the test specimen. The specimen is centered in the reduced section. The cross-sectional area of the measured part is reduced compared to the cross-sectional area of the remainder. It is important that the distance between the ends of the measuring part and the arms is large enough that the larger ends do not limit deformation.

The length of the transition area (shoulder) should be at least as large as the thinnest part. The total length of the reduced part should be at least four times longer than the maximal cross-section dimension. The length of the measuring part should be proportionally longer to its section. Otherwise, the state of tension will be more complex than simple tension [20,21].

#### 4.1.1. Method of Test Specimen Preparation

For standard tests, the prescribed number of samples must be observed. The standard currently specifies testing at least five test specimens. In the case of experimental work, this quantity is not defined. Dog-bone-shaped samples were cast from each mixture. One week after casting they were removed from the mold. The test specimens in which hydration took place hardened and solidified for 28 days. The preparation of the small-scale sample consisted of making a template and then preparing the forms (Figure 4).

The template was created based on a predetermined design. The geometric design considers all the principles that need to be respected. The basis of the mold consisted of a wooden base, plastic mass and a stainless-steel element for the specimen-gripping hole (Figure 5).

A transparent overlay film was stretched over the wooden substrate; this is for better post-hardening manipulation of the specimen. The plastic mass was adhered to the wooden base around the perimeter of the template. It was aligned at a height of 12 mm. The stainless-steel element was inserted inside after the template was removed and secured against displacement. Improper gripping of the specimen causes inaccurate test results, and therefore, the position of this element is very important.

#### 4.1.2. Gripping Specimens into the Machine

As already mentioned, the biggest problem is gripping the test object. The direction of the load applied to the specimen must be identical to the longitudinal axis of the specimen. The grip must be firm enough to prevent slippage between the bolts and the stainless-steel element embedded in the specimen (Figure 5a). Incorrect gripping of the specimen generates transverse forces. These affect the resulting measurement values. Bending will cause premature failure of the specimen and incorrect determination of the modulus of elasticity. For accuracy, a method of gripping the specimen using special teardrop-shaped metal fixtures was chosen.

The standard gripping method was chosen for the waste PET specimens. Flat jaws were used for tensile tests up to 5 kN (Figure 5b). The jaws must exert sufficient lateral pressure to prevent slippage between its surface and the test specimen.

### 4.2. Cement–Concrete Specimens

A dry mix method was used to make the specimens. This consisted of two components: naturally quarried aggregates and cement. The volume of mixing water needed to produce 1 m^3^ of fresh concrete depends on different requirements. It is important to achieve complete hydration of the cement, to hydrate the surface of the aggregate, and to achieve the desired consistency of the fresh concrete. A total of 0.1 L of mixing water was used for every kilogram of dry mixture in our case. The strength of the concrete corresponds to class C20/25. The volume weight of the hardened concrete is 2100 kg/m^3^. The individual components were dosed by weight with an accuracy of ±0.5%. In a prepared plastic container, the dry mix and the required amount of mixing water were mixed using an agitator. A homogeneous mixture was formed to be ready for use in the specimens. Mixing of the mixture from the time of the first addition of water should take at least 5 min. The sufficiently mixed mixture was gradually added in layers to the prepared form.

Each layer of concrete was gradually compacted. The mixture, thus prepared and sufficiently compacted, was left to work harden. After 7 days, the test specimens were removed (Figure 6).

### 4.3. Vinylester and Polyester Specimens

The chemical anchor PE 300 SF was required to make the polyester specimens. Due to its good mechanical properties, the tested specimen was reduced several times. Its shape was cut into cardboard paper. It was important to ensure even mixing of the two basic components of the sample. The sufficiently mixed mixture was poured into the form and aligned along the entire edge. The samples thus prepared were placed in a favorable location. Figure 7 shows the final specimens after the hardening and debonding process.

The preparation of vinylester samples differs in the composition of the material used. A vinylester-resin-based chemical anchor was used instead of a polyester-resin-based chemical anchor. This material has better mechanical properties compared to the polyester based chemical anchor.

### 4.4. Polyester Resin Specimen

Polyester 109 is a solution of unsaturated polyester resin in styrene. It consists of two components: a polyester resin and an initiator. Due to the high strength of this material, the shape of the specimens was identical to that of the polyester and vinylester test specimens. The form was made up of cardboard and a transparent cover sheet. The foil was also applied on top to avoid the mixture flowing into the mold perimeter walls. This prevented significant deformation of the shape of the samples. Once the mold was prepared and the surface was controlled, the polyester was dosed with the initiator. The ratio of the individual components is given by the manufacturer. To the required amount of polyester, 2% of initiator was added with stirring (polyester/initiator ratio 100/2). This material is characterized by a fast hardening time (up to 15 min at 20 °C). Consequently, it was necessary to work with a quantity of mixture that could be incorporated over time. The mixture thus prepared was spread evenly into the mold. The curing time is a maximum of 24 h.

### 4.5. Polyethylene Terephthalate (PET) Specimens

Specimens from waste PET bottles were used to test recycled polyethylene terephthalate (Figure 8). The samples were dog-bone shaped to avoid stress concentrations in the gripping part. By using this specimen shape, a simple tension in the specimen’s shank is recorded. Both ends of the specimens were gripped in the jaws. Jaws for tensile tests up to 5 kN were used. This type of jaw is directly designed for testing plastics.

### 4.6. Composite

Composite materials are made up of two or more chemically different components. According to the type of material used (thermoset), the composite in this experiment can be divided into two groups: composite using vinylester and composite using polyester resin. In total, several test specimens were created. The individual specimens differed from each other in formulation. PET fibers are an important component of the composite. In this experiment, waste PET bottles were used. The plastic bottles were cut into thin fibers to obtain continuous fibers (Figure 9a,b).

Straight plastic fibers usually have a low bond strength with concrete. The dispersion of the fibers in the specimen is critical to the tensile response of any type of fiber concrete. Therefore, it is necessary to ensure a constant casting process to achieve good mechanical properties and to ensure minimum dispersion of results. The specimens’ shape requires an optimal arrangement of fibers both on the shank and in the end zones.

In the experiment, test specimens were fabricated using PET fibers with a dimension of 5 mm × 15 mm. A fiber volume greater than 5% in concrete causes serious homogeneity and workability problems. Therefore, the result for this variant of test specimens was unsatisfactory. Thus, the use of smaller chips of 2 × 5 mm was resorted to (Figure 9c). Not only the fiber content, but also the fiber orientation, have a significant influence on the mechanical properties. This depends to a large extent on the geometry of the fiber and the method of manufacture. The PET fibers in this experiment are not straight. Their surface is slightly roughened due to the cutting technology, as shown in Figure 10. Within the volume of concrete, very different fiber orientations can occur locally.

At the beginning of each experiment, it is necessary to weigh all the components that were used in the composite mixture. The proportions of the mixture by weight are given in Table 1.

Reactoplast was first mixed into the prepared container (the preparation of vinylester and polyester resin has already been mentioned in this article). The reason for this is to ensure even mixing of the PET fibers with the reactoplast. Polymerization starts as soon as the two components of the material are combined. Polymerization is a poly-reaction in which macromolecules are formed by the fusion of molecules of a base substance (monomer) without the formation of by-products. It takes place in three successive stages. The first phase is initiation. This triggers a reaction linking the monomer units. In this phase of polymerization, recycled PET fibers were added to the mixture and the mixture was allowed to mix for 2 min (Figure 11).

It is necessary to work quickly due to the ongoing chemical reaction and rapid hardening. The processed mixture was poured in small volumes into a container containing the already prepared fresh concrete mixture. Mixing of all components was ensured by an agitator. This was used to achieve adequate workability and to maintain the required water/cement ratio. When all the ingredients were sufficiently mixed together, the composite began to be filled in small layers into the prepared mold. After each layer was placed, the concrete was compacted. The whole process took approximately 8 min. After 7 days of curing, the samples were removed from the mold. Due to the short time during the preparation of the test specimens, some of the mixture also got into the locations of the steel fixtures (Figure 12a,b). These parts were cleaned after hardening (28 days).

### 4.7. Experimental Measurement Process

The solution of each experimental task can be divided into three phases: preparation, measurement and evaluation. The dimensions of the test sample as well as the test procedure are defined by the standard. Test specimens have been designed to be gripped well in the test fixture and to generate no stresses at the gripping points. These could negatively affect the result of the experimental measurements.

The test was carried out on a SAUTER TVM-N test machine (Figure 13), which is designed for standard measurements with a force gauge of 5 kN at a constant loading rate of 1 mm/min and a speed accuracy of 3%. The test specimen is fixed on both sides in a joint formed by two smooth bolts and steel bands.

The screw passes through a stainless-steel hole in the sample. Double-sided grip is achieved in the test device by using an omnidirectional joint. This minimizes bending of the sample. The test device allows the sample to be loaded by a simple tension. The load is applied at a constant speed. The value of the force is recorded in the measuring notebook. Synchronously with the increase in force, the increase in the absolute elongation ΔL of the shank of the measured specimen at length *L* is also recorded. The standard test loading rate for is 1 mm/min. A single point on the specimen was fixed at the base, which will be supported by the type of the Sylvac deflection gauge. The ultimate strength of the material is derived from the maximum stress achieved in the tensile test.

The result of the test is a digital record of the force and the absolute deformation change. The data are recorded at a density of approximately 0.1 s. To evaluate the material properties, the relationship between the relative strain *ε_x_* and *σ_x_* must be interpreted graphically (stress–strain diagram). For this reason, all geometrical data (shape, weight, dimensions, etc.) were statistically processed for each specimen. The cross-sectional area A was calculated from the average dimensions of b and h. The normal stress *σ_x_* was then found using relation (1). The relative strain *ε_x_* is expressed by the accurately measured distance *L*. This parameter represents the distance between the fastening center point of the Sylvac gauge and the sensing tip of the indicator. This tip of the indicator leans on the fixed point on specimen. From the transformed values of *ε_x_* and *σ_x_*, a stress–strain diagram is displayed. The results of the experiments are presented in the following chapter.

For a better overview of the scientific work, a flow chart has been prepared for this experiment (Figure 14). The flow chart presents the most relevant activities that made up this experiment. The specific activities build on each other and are shown in sequential order.

## 5. Results of Measurements

This section presents the results obtained from tensile strength tests of cement–concrete, polyester, vinylester, polyethylene terephthalate, resin and composite specimens. Measurements on the specimens were taken up to the moment of rupture. For each specimen tested, deformation occurred in the narrower measuring section.

The tensile force and elongation were recorded for each of the specimens under the same conditions. From these data, the normal stress, strain and tangential elastic moduli were expressed. The calculations were based on Equations (1) and (2). The important geometrical parameters of the test specimens were measured with a caliper. Each test specimen was given a stress–strain diagram based on the processed data. A total of 16 valid measurements were taken.

### 5.1. Cement–Concrete Samples

For the evaluation of the measurement results on the cement–concrete specimens, the tangent to the curve of the stress–strain diagram was the most important. The curve starts at the point where the test line ended. It was caused by the reduced specimen size that was designed due to the small scale of the original test setup. For each of the samples, a deformation occurred in the shank of the test specimen (Figure 15).

From the data obtained, the stress–strain diagrams were processed, which are shown in graphs 1, 2, 3, and 4—(Figure 16).

In some cases, the measurements revealed that the specimen setting of the system at some point after the measuring device was started (Figure 5). Settling of the test device is related to the positioning of the sample and occurred in the stainless-steel section. At this point, a force drop occurred. Due to this phenomenon, the system was structurally aligned with the structural nonlinearities (slip, joint settlement, thread settlement, alignment to the equilibrium position). These samples were excluded from the statistical data set.

The actual analysis of the specimens focused on the tangential elastic modulus at the beginning of the branch (initiation elastic modulus), just before the failure limit, and the base elastic modulus. The measured and calculated elastic moduli are shown in Table 2.

The concrete used has a strength class C20/25 and a tensile strength of 1.5 MPa. For a better overview and comparison of the individual measurement results, a common stress–strain diagram was created for all the specimens (Figure 17). From the chart, the tensile strength can be read, which ranges from 1 to 1.15 MPa. These values do not match the values prescribed by the dry mix manufacturer. It is assumed that the differences may be mainly due to the degree of compaction of the fresh concrete. Another factor that influenced the final strength of the concrete is the fraction of aggregate used. The mix that was used contains aggregate with a small fraction, which is not suitable for use in load-bearing structures.

### 5.2. Reactoplastics

The next in the series of materials tested were reactoplasts. In order to describe the interaction between the PET fibers and the reactoplast with the cement–concrete component of the composite, the tensile properties of the polymers were investigated. The small-sized specimens tested were dog-bone shaped. During the tensile test, failure of the specimens occurred in the measuring part of the specimen (Figure 18). Table 3 shows the stress–strain diagrams of tensile tests of reactoplasts.

A nonlinear regression model of the polymers was developed for further analytical and numerical calculations (Table 4), (Figure 19).

### 5.3. PET

The test specimens from the waste PET were measured without specimens which had been notched during molding. These specimens had different locations of damage than their measuring part, which would have negatively affected the statistical analysis. The processed PET material results are presented in the graphs. In evaluating the tensile test of the PET material, we also considered the basic modulus of elasticity—Table 5.

Based on the data obtained, the measurements were compared with each other (Figure 20a). The tensile properties of the PET material could be evaluated using a constitutive model (hyperbolic regression), see Table 6 and Figure 21. A comparison of the regression curves of the waste PET material can be seen in Figure 20b. The small deviation between the measurements may have resulted from the carving process.

The evaluated experimental measurements were compared with the literature. After the comparison, the material properties of the waste PET are consistent.

### 5.4. Composite with the Addition of Vinylester

The composite specimens differed from each other not only in the composition and ratio of the individual components of the mixture but also in the color of the specimens themselves. Differences in the workability of the mixture and the associated consistency were observed when the specimens were prepared, which may have been mainly influenced by the proportion of the components and the preparation technology itself. PET fibers have low water absorption, so they cannot affect the hydration of the concrete. The hydrophobic property of PET fibers leads to the formation of purely mechanical bonds between the fibers and the cement matrix. The use of a combination of PET fibers and a vinylester anchor should provide better adhesion to the concrete mix. The stress–strain diagram of the specimen subjected to uniaxial tension is shown in Figure 22.

The evaluation of this composite can be divided into three phases, namely: the finishing of the system; the beginning of the interaction; and the conclusion, where the polymers act without cement–concrete. The first phase is probably influenced by the composition of the chemical anchor. The latter consists of a vinylester and an admixture. Therefore, this component of the composite has the characteristic of a reinforcing fiber. In the analysis of the material, where the creep of the system occurs, it is unnecessary to deal with the tangential modulus of elasticity, given that the material has reacted to a small extent. This section is primarily about searching for the equilibrium position of the specimen relative to the gripping mechanism. Once the specimen was set in the correct position, the interaction of the components of the composite occurred. The measured modulus of elasticity was 32.38 GPa. The linear regression was evaluated using mathematical software. When the cement–concrete could no longer withstand the maximum stresses, decomposition occurred and polymers themselves reacted. At that point, the tensile stress started to increase significantly and reached its maximum value (1.04 MPa). The inclination of the failure plane of the specimen was approximately orthogonal to the applied load. Measurements showed that the cement–concrete component did not interact with vinylester as predicted. This behavior of the individual components resulted in improved mechanical properties, especially in the modulus of elasticity. The modulus of elasticity before the failure limit was 78.82 GPa.

### 5.5. Composite with the Addition of Polyester Resin

The stress–strain diagram can again be divided into three parts (Figure 23). It is evident that the curves of the evaluated composites are different. For the composite using polyester resin, an improvement in the brittle base properties of the composite was recorded.

In contrast to composite 1, the composite using polyester resin had significantly better interaction of all elements. The modulus of elasticity in the part where all components interacted was 25.39 GPa. A higher tensile force caused the co-acting cement–concrete component and reactoplasticide to separate, leaving only the individual PET fibers to act. Finally, the loading stopped due to excessive deformation of the specimen, and the specimen failed (Figure 24).

The tensile stress reached its maximum value (1.01 MPa). The interesting thing about this type of stress–strain diagram is just the lower tensile strength, which was probably due to imperfect compaction. By an appropriate combination of material, ratio of components and technological procedure, we have succeeded in creating a composite that acts as a single homogeneous material.

Specimens of cement–concrete composite reinforced with PET fiber and reactoplasticide were subjected to tensile tests. Using a Keyence VHX 7000N digital microscope, microscopic images were taken at the failure point of the specimen. The microscopic images provide insight into the structure of the composite, information on the bond and adhesion between the matrix and the reinforcement material. The composite reinforced with vinylester resin and PET fibers can be seen in Figure 25. The magnification of the image is 2000 times. As can be seen, the interfacial interaction between the cement matrix and PET fibers was provided by the vinylester resin. The red circle in the microscopic images highlights that there are gaps between the matrix and the PET fibers. This was due to the lack of compaction, which was caused by the poor workability of the mixture.

## 6. Conclusions

The aim of this research work was to analyze and evaluate the behavior of the hybrid composite. The main idea of the work is to find out the mechanical properties of the investigated composite and its subsequent possible use in construction practice.

The measurements were carried out on the premises of the University of Zilina on the SAUTER TVM-500N testing equipment. In total, several tests were performed, but only 16 measurements were included in the statistical data set.

The advantage of these results is the independence of the stress–strain diagram (dependence of stress and strain) with the dimensions of the sample. For this reason, the design of scaled, long-necked, dog-bone specimens proceeded. This was also reflected in the resulting values, which, in some cases, reached thousandths of a millimeter. The result of the test was influenced by the material, geometry and the grip of the specimen itself. All the components included in the composite were tested; then, the composite itself was tested.

During the experimental phase, recycling PET fibers were mixed with one of several selected types of reactive plastics to create a hybrid composite. They were then added to concrete to improve the material and structural performance of road construction. Road structures are cyclically stressed and undergo stress alternation, which is associated with fatigue phenomena. This experiment is a basic input for further research. The ecological and efficient use of this waste material are other main motivations for this work.

The evaluation of the cement–concrete samples concluded that the resulting variations between measurements could be due to insufficient compaction rates. This was subsequently reflected in the resulting tensile strength, which ranged from 1 to 1.15 MPa. The purpose of this experiment was not to investigate the concrete itself, but its interaction with the other components of the composite. In analyzing the results, an exceptional phenomenon was found to occur with the setting of the test line. As it turned out, the sample started to strain after the testing device was started and eventually settled. The settlement of the test rig was related to the position of the sample and occurred in the part of the stainless-steel jig location. This caused the force to drop and the specimen to flatten. As a result of this phenomenon, the system structurally relaxed and only then started to react. Specimens in which this phenomenon was observed were excluded from the evaluation.

For another one of the materials tested, the research was concerned with investigating the behavior of selected polymers in simple tension. Three types of reactoplastics (polyester, vinylester, polyester resin) and one thermoplastic (PET) were selected. Due to the good mechanical properties of these materials, the size of the test sample had to be reduced several times. The experiments resulted in material characteristics under tensile stress. The properties of the individual components of the composite were investigated to describe their interactions in the composite, i.e., the participation of the PET fibers and the reactoplast with the cement–concrete component of the composite. Both reactoplasts and thermoplastics exhibit strongly nonlinear material behavior. In this work, several stress–strain diagrams were presented as results of tensile testing on test specimens. For further analytical and numerical calculations, nonlinear regression models of the polymers were also developed based on the assumption of their shape taken from the literature. All material property results were compared with those published in the literature, and it can be concluded that the individual material properties are in good agreement.

At the end of the experiment, the composite itself was tested. Two different variants of the composites were tested. The results obtained from this experiment indicate that the workability of the concrete mix is largely influenced by the size and shape of the PET fibers. Polymerization and hydration took place in the samples, which could have a great influence on their material properties. The evaluation of the measurements of the composite samples would not be objective because the results were largely influenced by the degree of compaction. Interestingly, with a lower quality of compaction, it is possible to achieve relatively high elastic moduli (see Section 5). The measured tensile strength was lower than that of the reference concrete. The stress–strain diagrams had a non-standard curve shape for both composites. This does not mean that the materials themselves behaved differently than expected; on the contrary, the physical behavior of the materials was confirmed. When evaluating the measurements, the different results of the interaction of the materials are interesting [21,22,23].

In conclusion, it can be stated that with the correct ratio of individual components, it is possible to produce a high-quality composite containing waste PET fiber. This can be used for structural pavement layers. However, further research is needed. The durability and rheology of the material should be addressed. For the continuation of this experiment, the design and construction of a larger geometric specimen must also be addressed. This involves the design of the gripping part, the shape of the fibers and the sample preparation technology itself. By changing the preparation technology, we can achieve better mechanical properties of the composite. The main idea of the new technology is to mix monomers, PET fibers and a cement–concrete mix. By mixing these components, hydration starts to take place, because an initiator is added to the whole mixture; this triggers another chemical reaction (polymerization). These changes can lead to better material properties of the composite. Further research will establish the suitability of this type of composite in road construction, which is the main purpose of its use [24,25,26].

Several modern case studies also investigate other types of composites in different materials composition and obtain relatively similar results. These authors take recycled polymers, cement–concrete and components used in civil engineering into account [27,28,29,30,31].

The main scientific research question of the investigated hybrid composites is to determine their mechanical properties. There is a clear description of material behavior under tension in the results chapter. This type of material has significant tensile stress due to the main component, cementitious cement, which becomes brittle after curing. It is generally acknowledged that plastic increases mechanical resistance dactylically.

### Current Research and Recommendations

The hybrid composite under investigation is currently being investigated for its compressive and flexural properties. Methods for processing PET particles and analyzing their shapes are also being optimized. Particle geometry will influence the mechanical properties of the composite. The number of composite variations is increasing due to the use of new freely available polymer bases. Further research is being carried out on the dynamic properties of the materials and, in particular, material attenuation.

## Figures and Tables

**Figure 1 polymers-15-03407-f001:**
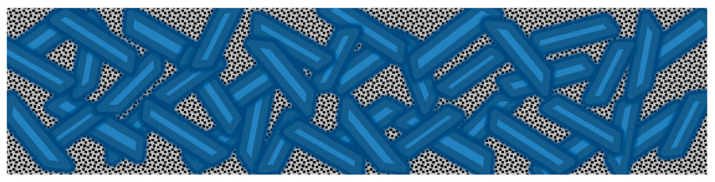
Hybrid Composite Composition Diagram. Legend: 
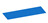
 PolyEthylene Tetraphthalate (PET)—reinforcing fibers; 
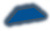
 Vinylester cover; 
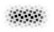
 Concrete with fine fraction of aggregate C20/25.

**Figure 2 polymers-15-03407-f002:**
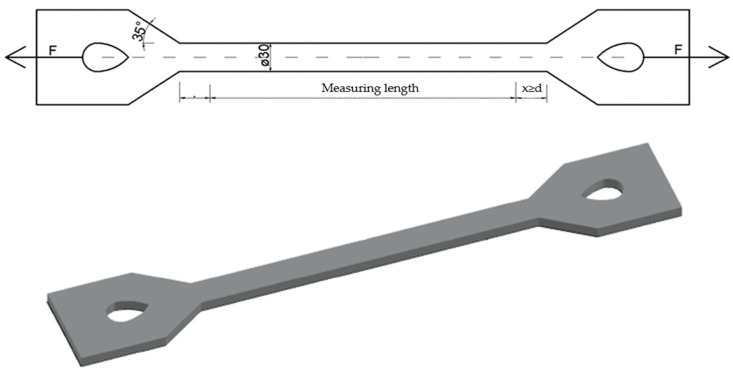
Test specimen model and 3D model of the test specimen used for the uniaxial tensile test.

**Figure 3 polymers-15-03407-f003:**
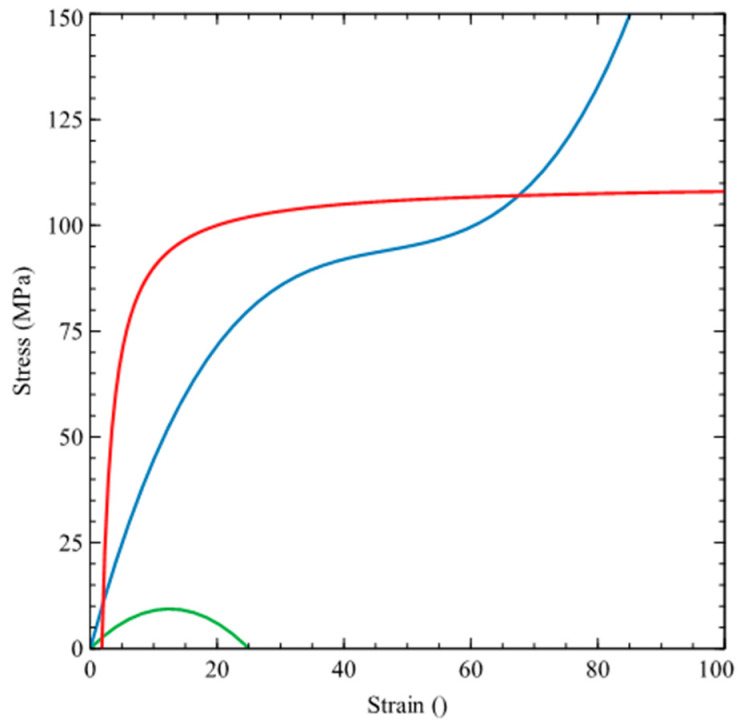
Predicted shapes of regression curves for individual materials.

**Figure 4 polymers-15-03407-f004:**
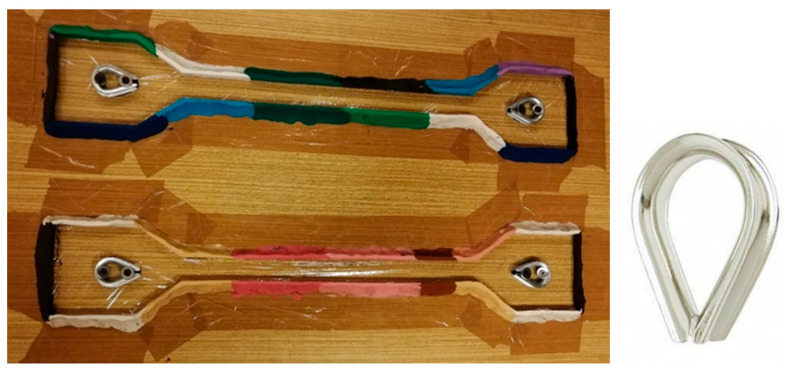
Form for the test specimens and a stainless-steel element for specimen-gripping hole.

**Figure 5 polymers-15-03407-f005:**
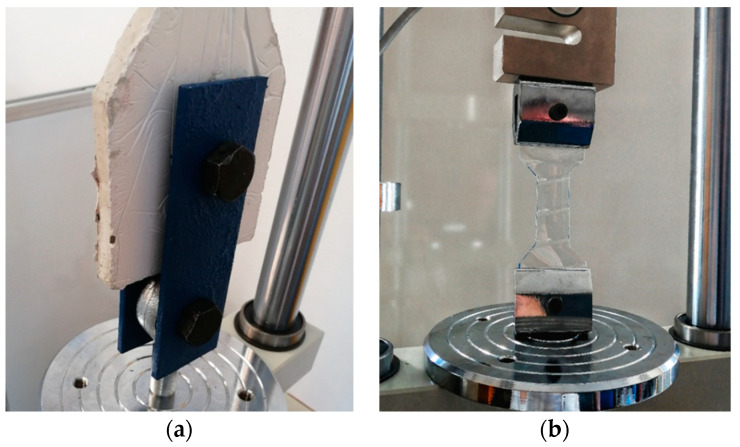
(**a**) Detail of gripping of the cast test specimen tested for single tension. (**b**) Gripping of the PET specimens in the test equipment.

**Figure 6 polymers-15-03407-f006:**
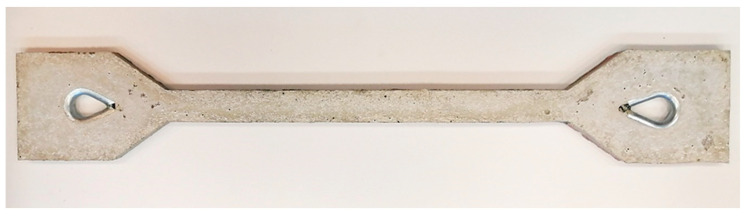
Prepared cement–concrete test specimen.

**Figure 7 polymers-15-03407-f007:**
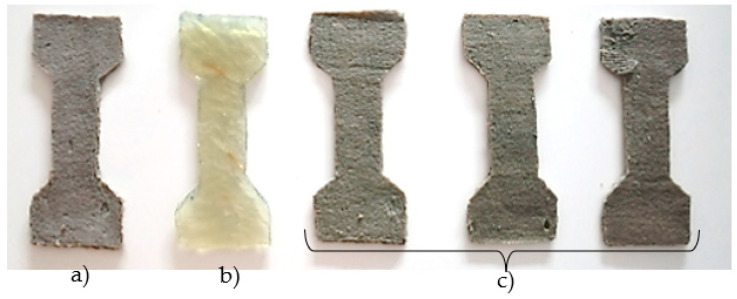
The final specimens of (**a**) vinylester, (**b**) polyester resin, (**c**) polyester.

**Figure 8 polymers-15-03407-f008:**
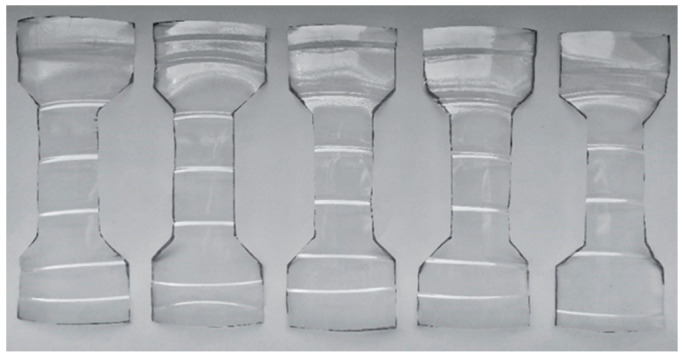
Polyethylene terephthalate (PET) specimens.

**Figure 9 polymers-15-03407-f009:**
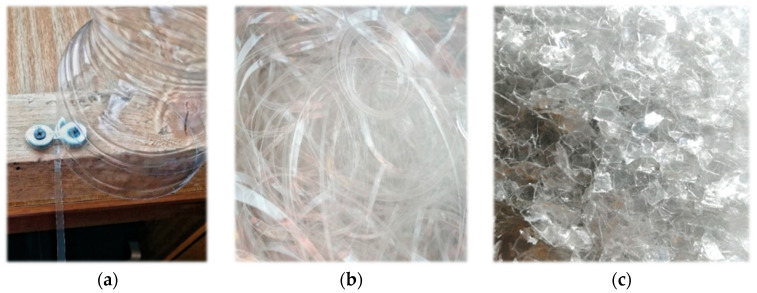
(**a**) Mechanism for uniform cutting of PET bottles. (**b**) Uniformly cut PET fibers. (**c**) PET chips size 2 mm × 5 mm.

**Figure 10 polymers-15-03407-f010:**
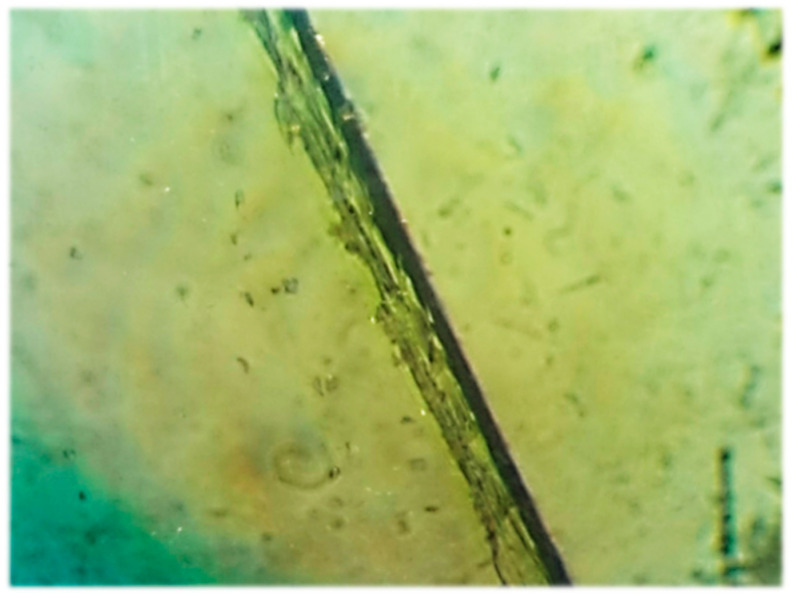
Cut PET fiber from a waste bottle under a microscope.

**Figure 11 polymers-15-03407-f011:**
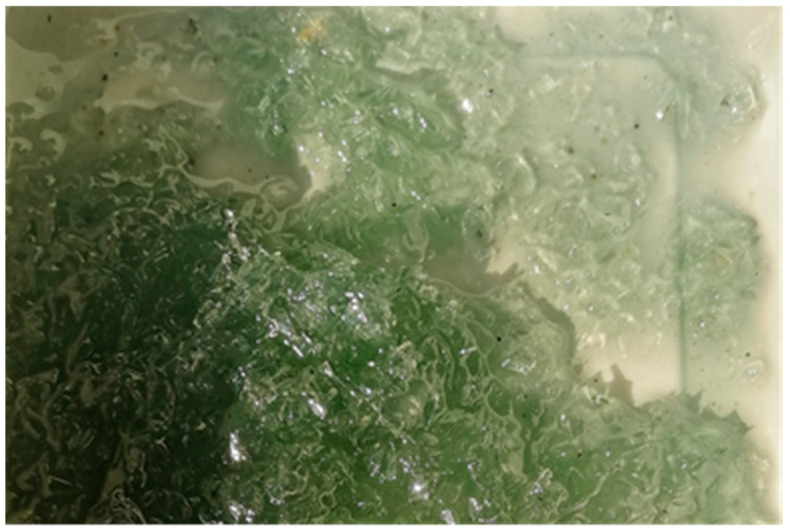
Polyester resin with PET fibers.

**Figure 12 polymers-15-03407-f012:**
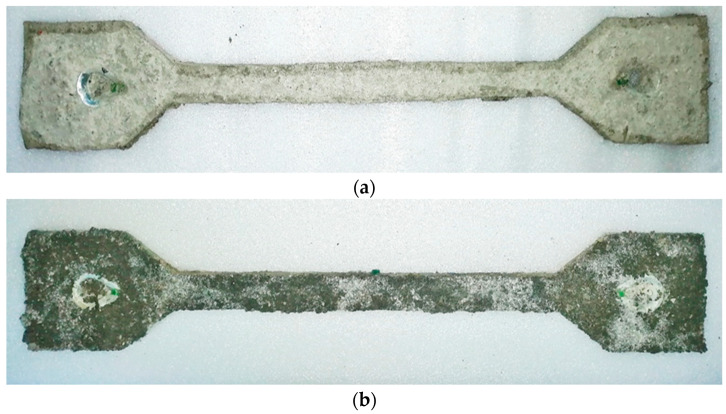
(**a**) Composite test specimen using a vinylester anchoring material. (**b**) Composite test specimen using a polyester resin.

**Figure 13 polymers-15-03407-f013:**
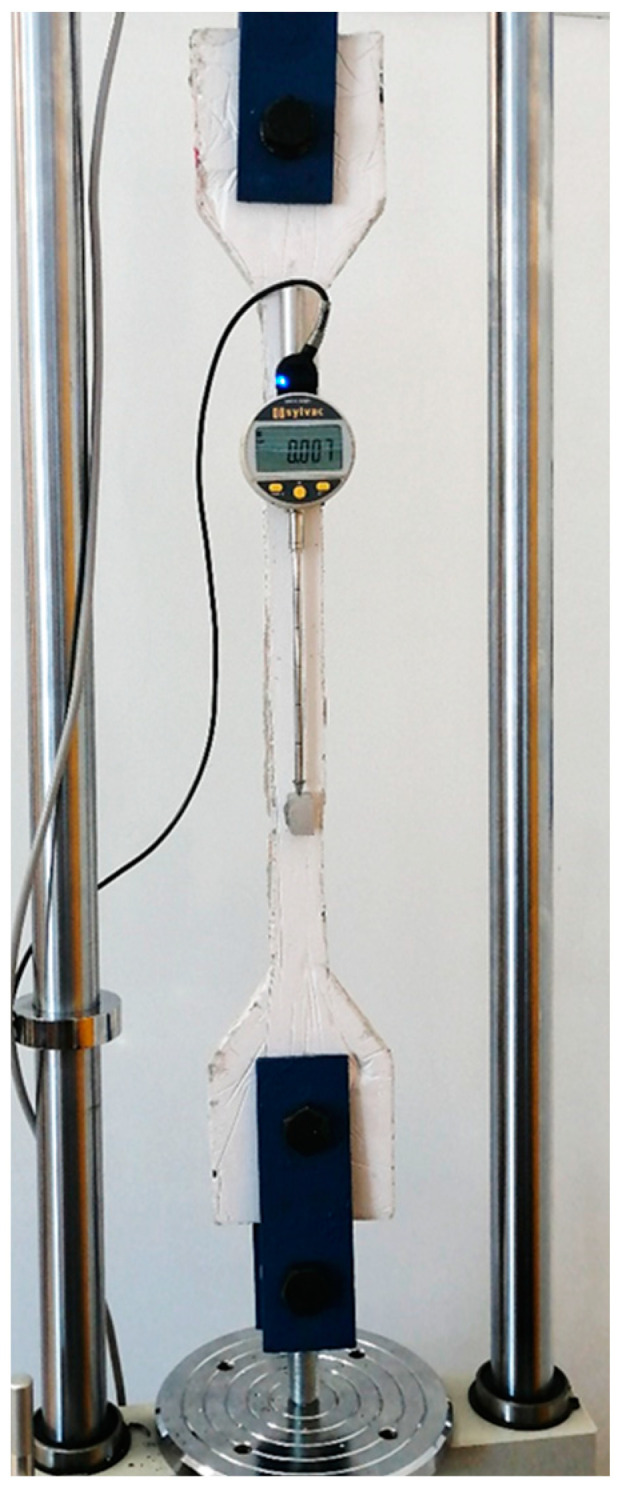
SAUTER TVM-N testing machine and specimen fixing before measurement.

**Figure 14 polymers-15-03407-f014:**
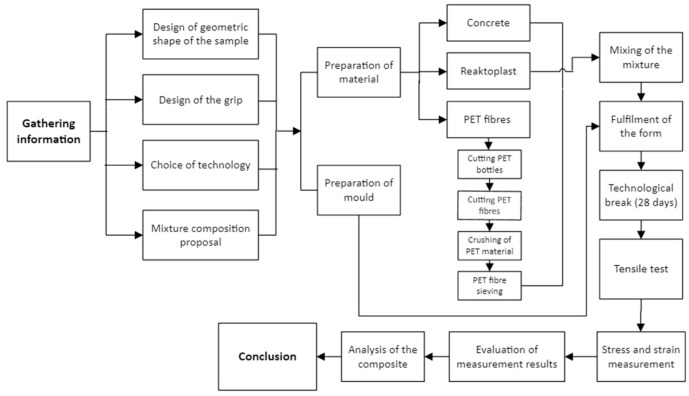
Flow diagram of the sequence of activities in the preparation of the hybrid composite.

**Figure 15 polymers-15-03407-f015:**
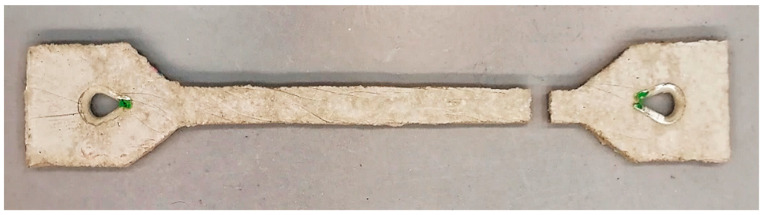
Cement–concrete specimen after rupture.

**Figure 16 polymers-15-03407-f016:**
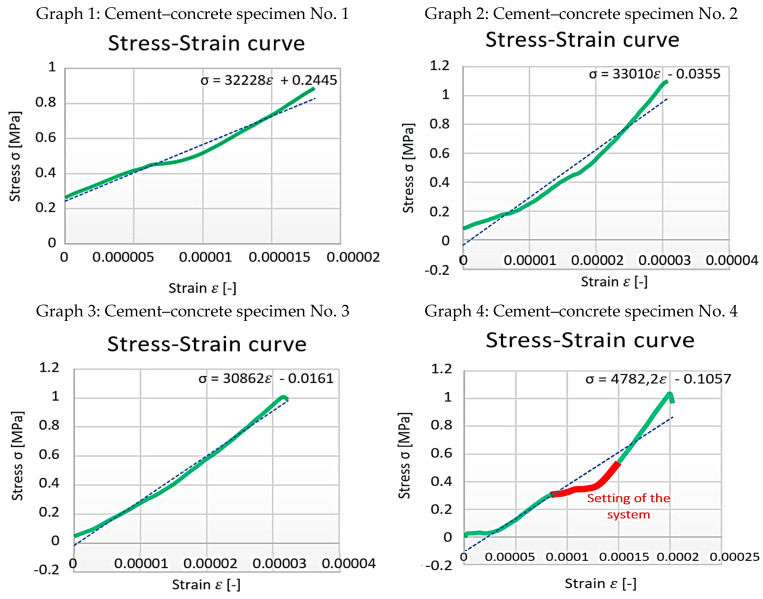
Cement–concrete specimen stress-strain diagram parts (green line) and the linear regression.

**Figure 17 polymers-15-03407-f017:**
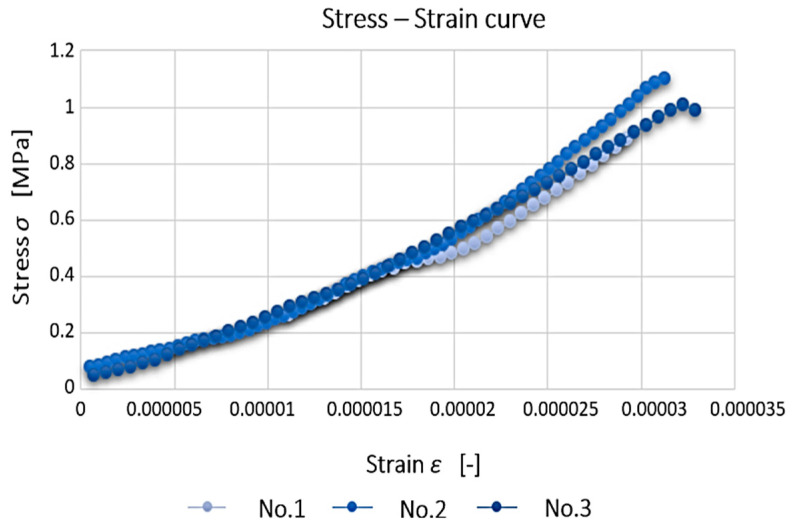
Comparison of results of cement–concrete specimens.

**Figure 18 polymers-15-03407-f018:**
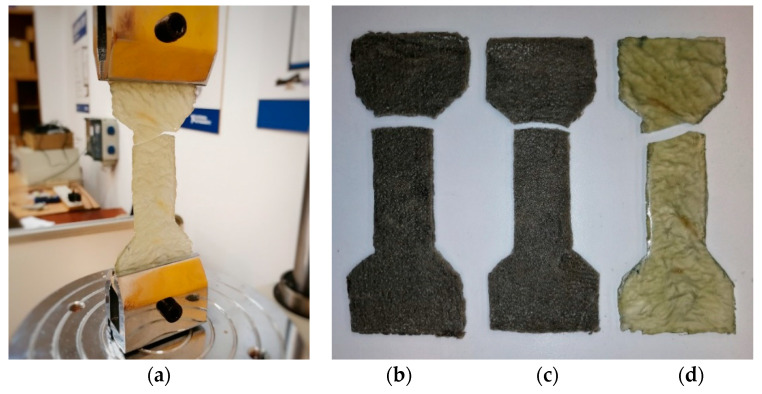
(**a**) Resin specimen in the machine and its breakage. (**b**) Breakage of the polyester specimen. (**c**) Breakage of the vinylester specimen. (**d**) Breakage of the resin specimen.

**Figure 19 polymers-15-03407-f019:**
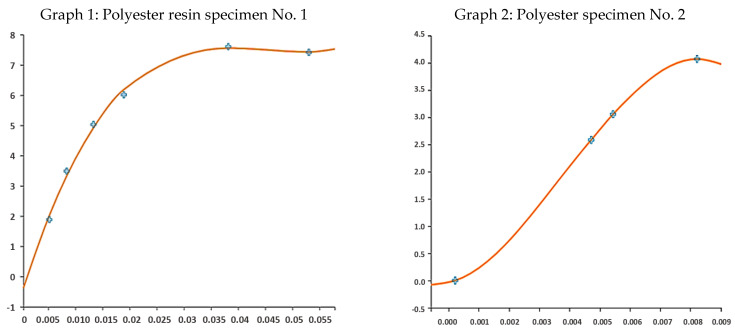
Regression curves of the specimens’ experimental result shapes (blue cross—measured data).

**Figure 20 polymers-15-03407-f020:**
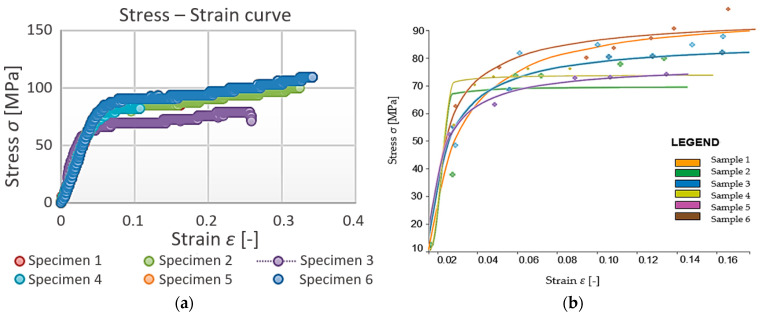
A comparison of specimen’s results and their regression curves of waste PET material. (**a**) original measured data, (**b**) hyperbolic regression of the measured data.

**Figure 21 polymers-15-03407-f021:**
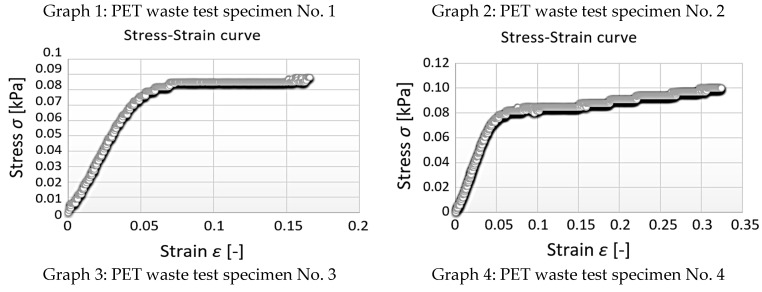
Individual stress-strain diagrams of 6 investigated specimens of PET waste material.

**Figure 22 polymers-15-03407-f022:**
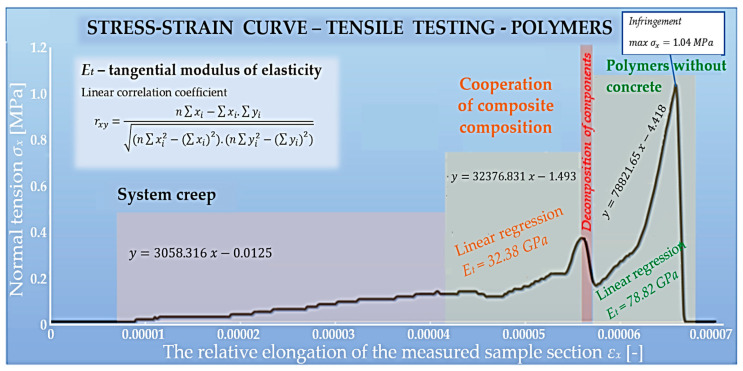
Evaluation of PET fibers and vinylester anchor composite.

**Figure 23 polymers-15-03407-f023:**
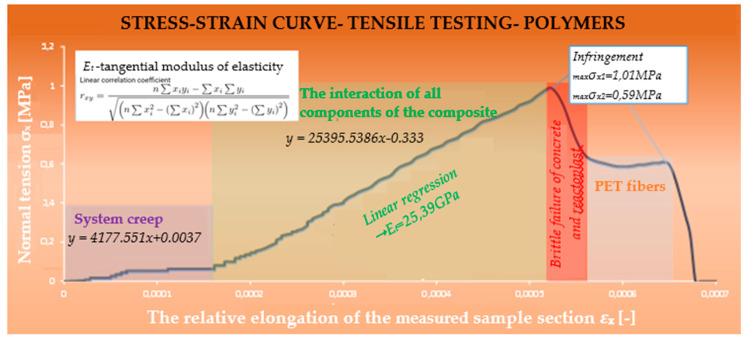
Evaluation of polyester resin composite.

**Figure 24 polymers-15-03407-f024:**
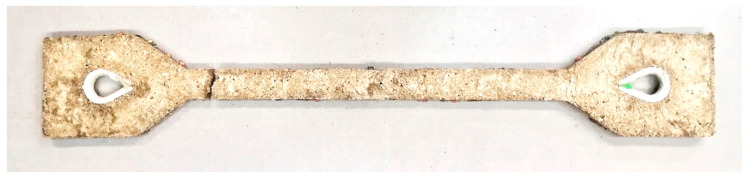
Polyester resin composite specimen after rupture.

**Figure 25 polymers-15-03407-f025:**
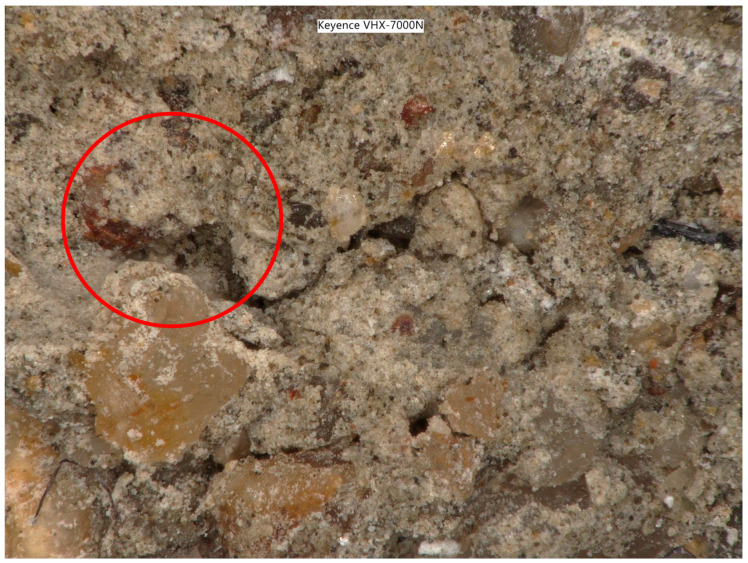
Microphotography at the point of failure of a vinylester containing composite-red circle.

**Table 1 polymers-15-03407-t001:** The proportion of individual components in the composite.

Specimens	Composite	Composition			
		Cement–Concrete Mixture [g]	PET Fibers < 2 × 5 mm [g]	Reactoplast [g]	Water [mL]
1	Polyester resin	1000	20	130	200
2	Vinylester	1000	20	50	200
3	Polyester resin	1000	10	130	200
4	Polyester	1000	20	30	200

**Table 2 polymers-15-03407-t002:** Modulus of elasticity of cement–concrete specimens.

Samples		1.	2.	3.
Modulus of elasticity [MPa]	At the beginning	31,501	23,678	25,177
	Before infringement	46,849	53,599	38,949
	Basic	32,228	33,010	30,862

**Table 3 polymers-15-03407-t003:** The stress–strain diagrams of tensile tests of reactoplasts.

Graph 5: Polyester specimen No. 1	Graph 6: Polyester specimen No. 2
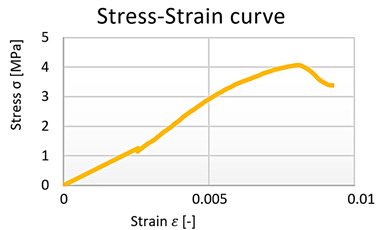	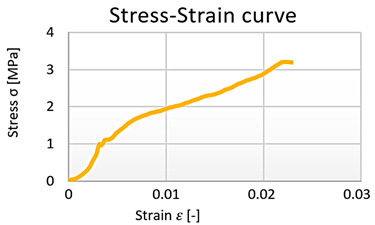
Graph 7: Polyester specimen No. 3	
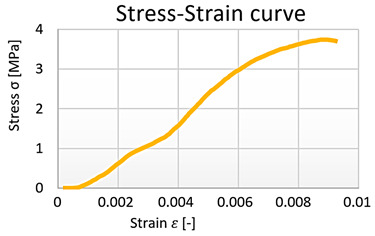	
Graph 8: Vinylester specimen	Graph 9: Polyester resin specimen
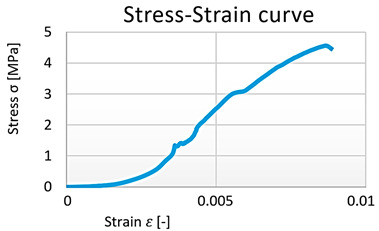	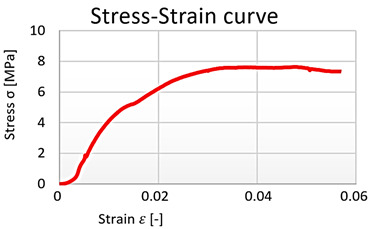

**Table 4 polymers-15-03407-t004:** Results of cubic regression (reactoplastics).

Material	Cubic Regression	Correlation Coefficient	Coefficient of Determination	Average Relative Error
Polyester resin	$y=-12,277,573.48x^{3}+14,923.38x^{2}+154.9022x-0.039$	1.0000	1	0.0000%
polyester	$y=98,321.2896x^{3}-13,171.5240x^{2}+573.8084x-0.6244$	0.9983	0.9965	2.9207%

**Table 5 polymers-15-03407-t005:** Modulus of elasticity of specimens from waste PET material.

Specimens	1.	2.	3.	4.	5.	6.
Modulus of elasticity [MPa]	1741	1860	3422	1591	1883	1711

**Table 6 polymers-15-03407-t006:** Results of hyperbolic regression of waste PET material.

Specimens	Hyperbolic Regression	Correlation Coefficient	Coefficient of Determination	Average Relative Error
1	y=97.6−1.4/x	0.9710	0.9429	3.8640%
2	y=78.4−0.6/x	0.9737	0.9480	2.3160%
3	y=78.4−0.6/x	0.9737	0.9480	2.3160%
4	y=91.8−1.0/x	0.9872	0.9746	6.1015%
5	y=106.0−1.6/x	0.9980	0.9959	1.3421%
6	y=106.9−1.6/x	0.9235	0.8952	2.7107%

## Data Availability

The data presented in this study are available on request from the corresponding author. At the time the project was carried out, there was no obligation to make the data publicly available.

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
