# Peer review of "Hybrid Composite Materials Made of Recycled PET and Standard Polymer Blends Used in Civil Engineering"

_polymers, 2023, doi:10.3390/polym15163407_

Round 1

Reviewer 1 Report

polymers-2254999
Title: "Hybrid composite materials made of recycled PET and standard polymer blends used in civil engineering"

The authors did a good job of studying the Hybrid composite materials made of recycled PET and standard polymer blends used in civil engineering; This scope needs such studies. However, in order to raise the quality of the manuscript to bring it to the required level in the journal “Polymers”, the reviewer recommends a deep processing of the comments (attached).

Author Response

The authors greatly value the work of the reviewer. Answers to valuable recommendations and questions can be found in the attached file.

Reviewer 2 Report

The authors tried to study the tensile properties of two composites made from recycled PET, cement concrete mix and standard polymer based adhesive mixes. The research idea is interesting, but authors failed to present the novelty of their work. In addition, manuscript was written with weak English. The manuscript suffers also some structural weakness and sections are not well linked.

Here are some comments:

§  Comment 1. The text needs an extensive proofread. The authors use words that are not English.

§  Comment 2: PET polymer should be somehow characterized, at least MFR should be given. 

§  Comment 3. In many cases an acronym is used without any definition. 

§  Comment 4: The dispersion of  PET fibers into matrix was not studied. SEM micrograph could give an idea on the matter.

§  Additional comments: other comments are presented on (polymers-2254999-review.pdf) file. 

Author Response

(The authors gave the same response as above.)

Round 2

Reviewer 1 Report

The manuscript has been well revised and can be processed for the next stage of publication.

Author Response

Thank you very much, for your valuable and helpful comments and suggestions.

Reviewer 2 Report

The authors globally addressed the mentioned issues. Language proofreading is needed before publication of paper (polymers-2254999).

Author Response

(The authors gave the same response as above.)
